# Magnitude, risk factors and economic impacts of diabetic emergencies in developing countries: A systematic review

**Halefom Kahsay Haile**[1]*, **Teferi Gedif Fenta**[2]

1 Department of Pharmacy, College of Medicine and Health Sciences, Adigrat University, Tigray, Ethiopia,
2 School of Pharmacy, College of Health Science, Addis Ababa University, Addis Ababa, Ethiopia

* heleka94@gmail.com

**Data Availability Statement:** All relevant data is within the manuscript and its supporting information files.

**Funding:** The author(s) received no specific funding for this work.

## Abstract

### Background

Diabetic ketoacidosis (DKA), hyperglycemic hyperosmolar syndrome (HHS) and severe hypoglycemia are considered as the life-threatening diabetic emergencies of diabetic patients worldwide. As the prevalence of diabetes grows in developing countries, so too does the impact of these costly human and economic complications. Noticeable scarcity of data concerning the magnitude, the cost expenditures as well as well unidentified predictors of these complications made the management more difficult in the resource limited health care settings. Thus, this systematic review aimed to assess the magnitude, risk factors and economic impacts of diabetes emergencies among diabetic patients in the developing countries.

### Methods

Following PRISMA (2020) guidelines, databases of PubMed, EMBASE, Cochrane and Scopus were searched for studies reporting on prevalence, risk factors, and direct costs of diabetes emergencies published in English from 2000 to 2023. Forty eligible studies were extracted and retrieved using manual data extraction form and automation tools. Studies were analyzed and combined in a narrative synthesis. The estimations of direct cost expenditure were standardized to 2023 USD.

### Result

A comprehensive examination was conducted on the 40 eligible studies, with the majority originating from African sources. The review shows the prevalence of diabetic emergencies; DKA episodes in the range of (3.8%-73.4%), HHS (0.9%-58%) and Severe hypoglycemia (3.3%-64.7%) per year in the developing countries. Infection, new onset of the diabetes, and non-compliance to medications and diets were reported as the most common risk factors of theses diabetic emergencies. Besides, the costs of hospitalization taken from the patients' perspective, that were associated per one diabetic emergency event per patient was reported in the range of 105–230 USD in the developing countries.

**Competing interests:** The authors have declared that no competing interests exist

## Conclusion

The rising prevalence of diabetic emergencies in poor nations, where infections, non-compliance, and new onset of diabetes are major causes, highlighted the urgent need for preventative interventions. Identifying high-risk individuals is crucial for implementing tailored strategies to reduce emergency visits and hospital admissions. The significant economic burden of these emergencies exacerbates the strain on already limited healthcare resources. In order to enhance health outcomes and lessen the financial strain on healthcare systems in these areas, preventive strategies must be incorporated into diabetes management programs.

## Introduction

The global incidence of diabetes is rising rapidly, posing a significant public health challenge, particularly in developing countries, which account for approximately 80% of the 537 million people living with diabetes worldwide as of 2021 [1]. According to predictions from the Global Burden of Disease studies, non-communicable diseases (NCDs) are expected to increase and surpass infectious diseases by 2030 [2]. About 643 million people worldwide are expected to have diabetes by 2030, most of whom live in poorer nations [1–3].

Diabetic Emergencies (DMEs), a serious health and economical concern, are one of the major obstacles faced by adults living with diabetes. It is characterized by abnormally high or low blood sugar levels, are significant contributors to morbidity, mortality, and increased healthcare costs among adults living with diabetes. For instance, diabetic ketoacidosis (DKA) and hyperosmolar hyperglycemic state (HHS) collectively account for up to 30% of diabetes-related hospital admissions in developing countries, with mortality rates ranging from 5% to 20% depending on the region and healthcare access [4, 5]. In developing nations, like India, around 1 million diabetes-related deaths occur annually, with acute complications such as DKA and hypoglycemia contributing significantly to these figures. the standard of care for diabetic emergencies is still insufficient, despite the disease's high prevalence and dangerous consequences [6–9]. Investment in chronic illnesses such as diabetes can potentially reduce the overall disease burden by 5–10%, according to the World Bank [10]. However, information about the costs of treating chronic conditions like diabetes in poor countries is scarce.

Emergency admissions resulting from acute metabolic crises in diabetes patients, particularly HHS and DKA, present formidable health challenges [11]. HHS, though less frequent and more common in Type 2 Diabetes (T2DM), is associated with higher case fatality rates ranging from 10–50% due to delayed diagnosis and treatment, significantly higher than DKA [12–17]. Reports from Nigeria and Ghana underscore the substantial contribution of diabetic emergencies to overall mortality [18–20].

Diabetic ketoacidosis (DKA), commonly associated with Type 1 Diabetes Mellitus (T1DM), is linked to several risk factors, including poor blood glucose management, alcohol consumption, and psychological stress. Studies have shown that inadequate insulin therapy and missed doses account for up to 40% of DKA episodes, while alcohol use and psychological factors, such as depression or eating disorders, further exacerbate the risk [12, 16, 21–29]. Severe hypoglycemia, another frequent diabetic emergency, is recognized as a significant barrier to insulin therapy and is associated with higher mortality rates [30, 31].

Diabetic emergencies also have considerable economic implications, with hospitalization costs accounting for a significant proportion of diabetes-related healthcare expenses. For example, the average cost of hospitalization for diabetic ketoacidosis (DKA) ranges from $1,500 to $5,000 per episode in low- and middle-income countries, imposing severe financial strain on individuals and families. As a large proportion of individuals with diabetes (approximately 50%–60%) fall within the economically active age group of 45–65 years. This not only reduces workforce productivity but also exacerbates economic strain by increasing healthcare expenditures and reducing household incomes due to medical costs and lost wages. Furthermore, the long-term complications associated with these emergencies contribute to increased healthcare system expenditures, often exceeding 20% of national diabetes care budgets in some developing countries [32–38].

The substantial direct and indirect medical costs associated with DKA and HHS are further exacerbated by the limited allocation of national health budgets to diabetes care. For instance, in many developing countries, less than 5% of the total health budget is dedicated to non-communicable diseases, including diabetes, leaving insufficient resources to manage acute complications like DKA and HHS effectively [39–47].

Even though diabetic emergencies are becoming more common in poor nations, there is a conspicuous visible scarcity of paucity of thorough in-depth studies in this area, and the research that is currently available lacks statistical clarity and specificity. To close this gap, a comprehensive evaluation is therefore required. In order to address this problem, this manuscript gathers and analyzes data from research carried out since 2000, concentrating on the economic effect, risk factors, and prevalence of diabetic emergencies among individuals who have been diagnosed with the disease. In doing so, this review hopes to offer important insights into the difficulties, management techniques, and preventative initiatives related to diabetic emergencies in developing countries.

## Materials and methods

A comprehensive systematic search of studies focused on the prevalence, risk factors and economic impacts of diabetic emergencies in developing countries were conducted between September and November 2023, following the Preferred Reporting Items for Systematic Review and Meta-Analyses (PRISMA) guidelines (S2 File), The protocol for this review has been registered in PROSPERO (ID: CRD42023494195). Ethical approval is not required as this systematic review will retrieve scientific literature available in public databases [48, 49].

### Eligibility criteria

Studies that used observational (cross-sectional, case-control, prospective and retrospective cohorts) epidemiological designs involving known diabetic patients in the developing countries setting were considered. For each publication retrieved from the electronic search, the titles and abstracts were initially assessed according to specific inclusion and exclusion criteria. Articles published from 2000 to 2023 in the English language with full text and reported original research findings from the developing country based on the 2023 world bank designations [50], that addressed one or more of the focused areas of the review were included. Costs presented in the studies under review are converted to 2023 international dollars prices. On the other hand, we excluded studies involving diabetic children below 5 years of age, as well as articles with unclear reporting that did not meet the defined criteria of the American Diabetic Association (ADA), JBDS, and AACE/ACE for Diabetic Ketoacidosis (DKA), Hyperglycemic Hyperosmolar (HHS), and severe hypoglycemia [51, 52].

## Data sources and search strategies

A search strategy was developed using key concepts in the research question: Diabetes Mullitus emergencies, acute complications, prevalence, Risk factors and economic burden, cost and developing countries. We searched PubMed, Cochrane, MEDLINE, EMBASE, SCOPUS for studies on prevalence, risk factors, and cost of diabetes emergencies published in English from 2000 to 2023. Other sources such as Google Scholar were hand-searched for additional studies and we included 40 studies for our analysis. In addition, references of included articles were hand-searched for any other original articles.

## Study selection

Titles and abstracts were examined for inclusion by two reviewers (HK and TG). Full copies of papers which appeared to fulfil the inclusion criteria were obtained and were independently selected by the reviewers for inclusion in either phase of the review. Disagreements were resolved by discussion.

At this second stage, publications in languages other than English, were excluded. Additionally, reviews, editorials, commentaries and opinion pieces (all non-peer-reviewed publications) were excluded. Lastly, papers that provided only abstracts or vague reporting of methods or results, non-rigorous sampling design, unsuitable outcome definitions or severe methodological limitations in their statistical analyses were also excluded. The full search protocol is available from the corresponding author on request. studies of fewer than 50 diabetic patients were excluded. This restriction was deemed reasonable given the epidemiological outcomes of interest (prevalence rate), as deriving these values from a very small patient population would lead to a high degree of uncertainty in the estimates. We excluded studies if: 1) they were pharmacological trials or the study methods involved any alteration to a participant's treatment or care, either pharmacological or behavioral; 2) articles with majority of participants were pregnant, fasting, on a restrictive diet, or were selected on the basis of having a specific acute or chronic illness;

## Quality assessment

We employed specific assessment tools tailored to each study design to evaluate the quality of individual studies. For cross-sectional studies, we utilized the Joanna Briggs Institute (JBI) evidence synthesis critical appraisal checklist, as recommended by Moola S. and colleagues [53]. Additionally, for cohort studies, the Newcastle-Ottawa Quality Assessment Scale, as advocated by Wells G., was applied [54]. Furthermore, studies focusing on the evaluation of cost expenditures related to diabetic emergencies underwent assessment using the British Medical Journal economic evaluation checklist developed by Drummond [55]. Any disagreements in the assessment process were resolved through discussion.

## Data extraction

Manual data extraction form as well as Covidence and EPPI-Reviewer automation tools were utilized and pilot tested, with adaptations made accordingly. For every study, the fist reviewer (HK) separately retrieved the following information when data were available: Data for collection includes title, authors, study design, publication year and country, sample size, participant type of DM and major findings of the interested outcomes. Any disagreements regarding data extraction were resolved through discussions with a second reviewer. Then extracted studies checked for accuracy by a senior researcher (TG) for final inclusion of the studies. Disagreements were resolved through discussion and with reference to the original article. Missing

data in the included studies were addressed systematically. During data extraction, studies with incomplete or unclear information on key variables were identified. Corresponding authors were contacted to request the missing data. If data remained unavailable, the studies were included but flagged for incomplete reporting, and their limitations were noted. Besides, missing numerical data were estimated using related variables or comparable studies, with methods clearly documented. The potential impact of missing data was well assessed and all efforts to handle missing data were transparently reported.

## Data analysis

A meta-analysis was not possible due to the heterogenicity of the include articles in study design, populations and outcome measures as well as incomplete report of the key outcomes. Therefore, Studies were combined in a narrative synthesis. Possible reasons for conflicting results were also reported narratively.

## Descriptions of the interested outcomes

This systemic review includes the studies and reports that describe the diabetic emergencies as based on international code of disease (ICD) for acute complication of diabetes mellitus. Particularly, **Diabetic ketoacidosis (DKA)** was defined based on a clinical diagnosis of DKA when there is random blood glucose level (RBGL) of $> 250$ mg/dl, urine ketone body of $\geq +2$, Arterial pH of $< 7.3$, and Bicarbonate of $< 15$ meq/l. **Hyperglycemic Hyperosmolar Syndrome** (HHS was considered when RBGL is $> 600$ mg/dl, with alteration in mental status with minimal or absent urine ketone body in diabetic patients.

**Severe Hypoglycemia**: described as a diabetic patient required assistance of another person and visited emergency health care settings for management.

**Cost:** was defined as the total direct costs (expressed in 2023 USD) expensed for the diagnosis, medical treatment, and hospitalizations associated with the diabetic emergencies (DKA, HHS and severe hypoglycemia) of the diabetic patients.

## Results

The primary search strategy identified 962 articles, with 392 duplicates subsequently excluded. Out of the remaining 570 articles, 105 were eliminated due to reasons for diabetic emergencies were associated with some expected factors. Additionally, 179 articles were removed during the initial title screening due to lack of information, and 161 were excluded during the second level of abstract screening because of irrelevant conte of the interested outcomes, resulting in 103 articles for full-text evaluation. Among these 125 studies, 65 were excluded for various reasons, including non-English language, conference presentations, unavailability of the full text, lack of peer review, or being a literature review on diabetes. Finally, we include 40 studies that presented results on prevalence, economic impact and risk factors of diabetic emergencies (Fig 1).

## Magnitude of diabetic emergencies

The results were clearly presented in most studies and consistently well reported in relation to the methods adopted (Table 1). It summarizes the methodology and key conclusions of the 26 Articles that were systematically reviewed the prevalence and incidence data to evaluate the magnitude of diabetic emergencies. It is important to note that the populations studied and the denominators used for calculating prevalence differ among the studies. Therefore, the prevalence rates presented in this review should not be directly compared across studies. As

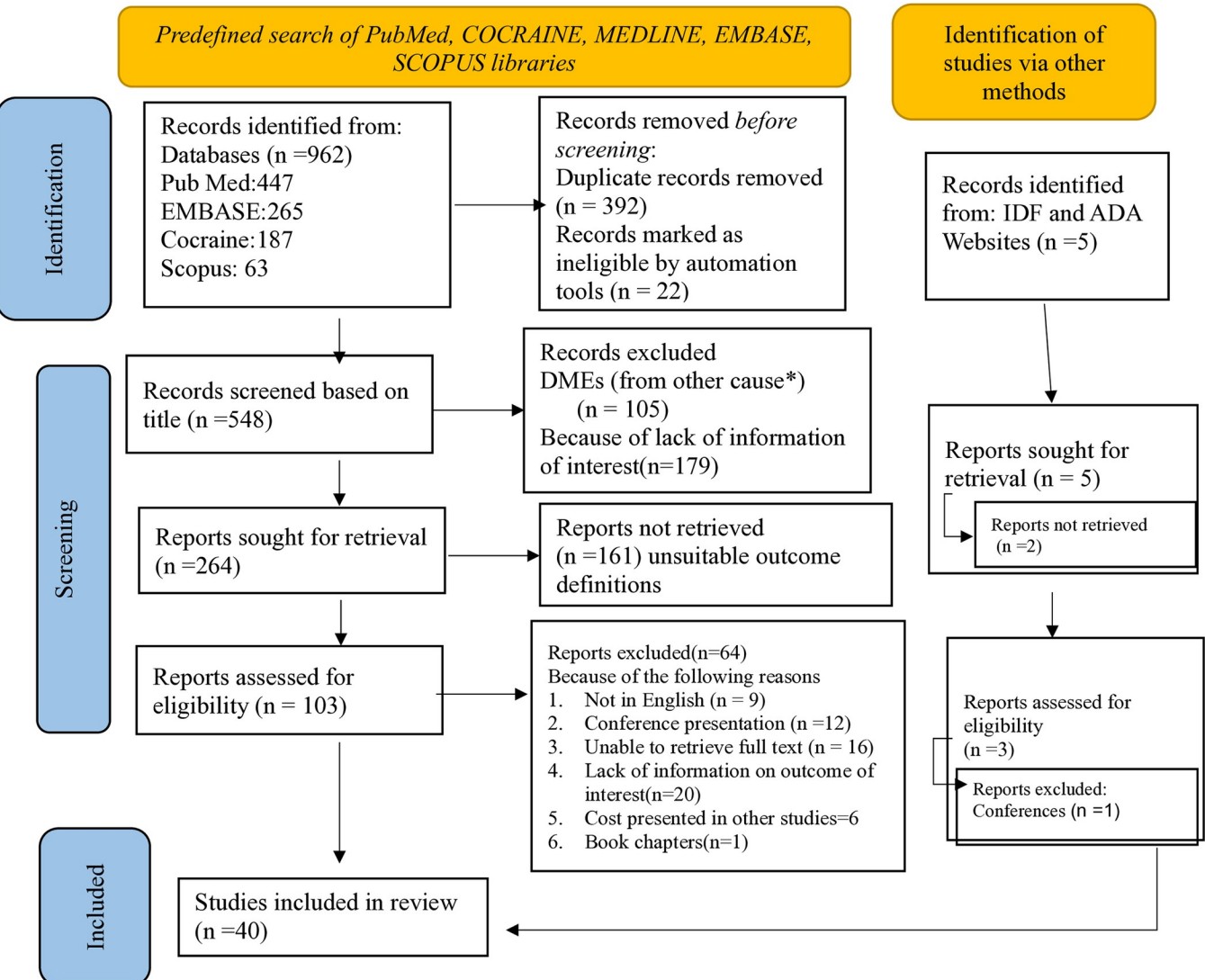

*Others cause; severe malaria, severe malnutrition and Gestational pregnancy*

**Fig 1. Flow diagram for selection of studies.** This flow chart illustrates the process of identification, screening, and inclusion for studies in the systemic review.

presented in (Fig 2), majority of the studies (n = 20) were carried out in Africa [18–20, 56–72] with the remaining five papers coming from Asian nations such as China, Saudi Arabia, India, and Iraq [73–77]. One article was also included from Mexico [78]. Five researches provided a combined report on the prevalence of hyperglycemic crises (DKA and HHS) and almost half of the studies evaluated the prevalence of each of the three diabetic emergencies separately [18, 19, 62, 63, 65]. The articles comprised a total of (n = 19,310) participants who were aware of having diabetes. Overly, four studies fully reported prevalence of diabetic emergencies [20, 56, 61, 66] and the remaining studies reported prevalence of DKA and HHS as a hypergly-cemic crisis as well as hypoglycemia is reported separately from one study [72]. Studies that carried out in Uganda and Saudi Arabia have documented a range of prevalence rates for DKA episodes, from 6.7% to 83%, respectively [64, 77]. Notably, Uganda documented the

**Table 1. Characteristics of included studies in qualitative analysis for magnitude of diabetic emergencies.**

| Authors | year | Name of Developing countries | Study type | DMEs Sample size (n) | Population (DM type) T1 (T1DM) T2 (T2DM) | Magnitude of diabetic emergencies (%)[a] | | | Outcomes Data type (Prevalence/ Incidence) | Definitions of Diabetic emergencies |
|---|---|---|---|---|---|---|---|---|---|---|
| | | | | | | DKA | HHS | SH | | |
| [56] Lotter N. et al., | 2021 | Captown, S. Africa R | 24-week Retrospective | 197 | T1 and T2 | 48.7 | 6 | 22.3 | Prevalence | **DKA** was defined as hyperglycemia with glucose >13.9 mmol/l, metabolic acidosis with pH <7.3 and bicarbonate <18 mmol/l, and presence of ketonemia (>3 mmol/l) **HHS** was defined as severe hyperglycemia (serum glucose >33.3 mmol/L), hyperosmolality (serum osmolality >320 mOsm/kg), marked dehydration and the absence of significant acidosis (pH > 7.3, bicarbonate >15 mEq/L); ketonuria may be slight or absent. SH was defined as a blood glucose level <3.9 mmol/l with an altered level of consciousness. |
| [57] Jasper, et al., | 2014 | Jos, Nigeria | Retrospective | 2470 | T1 and T2 | 12.2 | | 8 | Prevalence | ICD Codes |
| [58] Tilaye et al., | 2021 | Adama, Ethiopia | Cross-sectional | 200 | T1 and T2 | 66.5 | 10.5 | Prevalence | DKA was defined as a potentially life-threatening complication of DM characterized by absolute insulin deficiency and HHS was defined based on ICD code | |
| [73] Kasinathan D. et al., | 2013 | Sivaganga, India | Cross-sectional | 284 | T1and T2 | 53 | | | Prevalence | ICD code |
| [59] Edo A E., | 2012 | Legos, Nigeria | Registry | 84 | T1 and T2 | 42 | 58 | | Prevalence | DKA: was defined as glucose >250 mg/dL), serum ketonemia and/or ketonuria, and acidaemia (serum bicarbonate <18 mmol/L). HHS: was defined by the presence of glucose>600mg/dL), hyperosmolarity >320 mOsmol/L with little or no ketonemia/ketonuria. |
| [60] Mbugua, P. K., et al., | 2005 | Nairobi, Kenya | Nine-month prospective study | 648 | T1 and T2 | 7.4 | | | Prevalence | ICD code |
| [61] Negera, et al., | 2020 | Jimma, Ethiopia | Cross-sectional | 348 | T1 and T2 | 73.4 | 22.8 | 3.3 | Prevalence | ICD codes |
| [62] Rudasingwa, GJ., et al., | 2012 | Kigali,Rwanda | Cross-sectional | 294 | T1 and T2 | 23 | | 16 | Prevalence | ICD codes |

(*Continued*)

**Table 1.** (Continued)

| Authors | year | Name of Developing countries | Study type | DMEs Sample size (n) | Population (DM type) T1 (T1DM) T2 (T2DM) | Magnitude of diabetic emergencies (%)[a] | | | Outcomes Data type (Prevalence/ Incidence) | Definitions of Diabetic emergencies |
|---|---|---|---|---|---|---|---|---|---|---|
| | | | | | | DKA | HHS | SH | | |
| [65] Ogbera, A. O. et al., | 2009 | Lagos, Nigeria | Prospective | 111 | T1 and T2 | 85 | | 15 | | DKA: referred to blood glucose levels >13.8 mmol/L and the presence of metabolic acidosis (bicarbonate levels of <10 mmol/L-18 mmol/L) and or the presence of ketonaemia or ketonuria. HHS: referred to plasma glucose levels of >33.3 mmol/L and bicarbonate levels of >18 mmolL with or without the presence of ketonuria |
| [63] Iradukunda A., et al., | 2021 | Rwanda | 5-Year Facility-Based Retrospective Study | 246 | T1 and T2 | 95 | | 4.1 | Prevalence | ICD codes |
| [64] Bateganya, M. H. et al., | 2003 | Mulago Uganda | Cross-sectional | 3103 | T1 and T2 | 6.7 | 3.3 | | Prevalence | ICD codes |
| [18] Ogbera, A. O. | 2007 | Nigeria | Prospective | 206 | T1 and T2 | 40 | | | Prevalence | DKA: Hyperglycemic emergencies with ketotic state HHS: hyperosmolar non-ketotic state |
| [66] Desse, T. A., et al., | 2015 | Jimma, Ethiopia | Retrospective | 421 | T1and T2 | 20.5 | 54 | 20.9 | Prevalence | DKA was defined as admission blood glucose >250 mg/l and urine dipstick ketone level ≥ +2 HHS was defined as blood glucose >600 mg/dL, alteration in mental status and mild or absent ketonuria. Hypoglycemia was defined as a blood glucose level <70 mg/dL |
| [20] Sarfo-Kantanka et al., | 2016 | Kumasi, Ghana | 31-years of retrospective | 8020 | T1 andT2 | 10.7 | 26.1 | 17.0 | Prevalence | ICD code |
| [74] Li J., et al., | 2014 | Guangzhou, China | Cross-sectional | 611 | T1 | 3.8 | | 16.2 | Incidence | DKA: blood plasma glucose >13.9 mmol/L, blood bicarbonate <15 mmol/L and/or pH < 7.30 (arterial) and elevated level of ketones in the urine or blood. SH: an event requiring assistance of another person to actively administer carbohydrate, glucagon or other resuscitative actions |
| [67] Gebre. B. B., et al., | 2019 | Gurage zone, Ethiopia | Cross-sectional | 338 | T1 and T2 | 14.2 | 6.2 | 6.5 | Prevalence | ICD codes |

(*Continued*)

**Table 1.** (Continued)

| Authors | year | Name of Developing countries | Study type | DMEs Sample size (n) | Population (DM type) T1 (T1DM) T2 (T2DM) | Magnitude of diabetic emergencies (%)[a] | | | Outcomes Data type (Prevalence/ Incidence) | Definitions of Diabetic emergencies |
|---|---|---|---|---|---|---|---|---|---|---|
| | | | | | | DKA | HHS | SH | | |
| [75] Al-Obaidi AH., et al., | 2019 | Basrah, Iraq | Cross-sectional | 147 | T1 | 49 | | | Prevalence | DKA was defined as a tetrad of blood glucose > 200 mg/dL (11 mmol/L), ketonemia and ketonuria, venous pH < 7.3 and/or bicarbonate < 15 mmol/L |
| [68] Adem A., et al., | 2011 | Addis Ababa, Ethiopia | 4-year retrospective study | 724 | T1 and T2 | 71.1 | | | Prevalence | ICD code |
| [69] Abate, M. D. et al., | 2023 | Bahrdar, Ethiopia | 5-year Multicenter retrospective | 453 | T1 and T2 | 12.5 (35.6 in T1 and 6.3 in T2) | 2.1 (0.9 in T1 and 2.4 in T2) | | Incidence | DKA: was defined based on a clinical diagnosis of DKA when there is RBGL of > 250 mg/dl, urine ketone body of ≥ +2, arterial pH of < 7.3, and Bicarbonate of < 15 meq/l. HHS was considered when RBGL is > 600 mg/dl, with alteration in mental status with minimal or absent urine ketone body |
| [78] Doubova SV et al., | 2018 | Mexico-city Mexico | Cross-sectional | 192 | T1 | 11.9 | | 8.3 | Incidence | ICD code |
| [19] Ajayi, E. A. and Ajayi, A. O. | 2009 | Ekiti, Nigeria | 5-year retrospective | 2,696 | T1 and T2 | 11.86 | | | Prevalence | ICD codes |
| [77] Ahmed MM. | 2014 | Jeddah S. Arabia | 1-year prospective study | 117 | T1 and T2 | 83 | 17 | | prevalence | DKA: referred to the presence of metabolic acidosis (bicarbonate levels of <10 mmol/L-18 mmol/L) and the presence of ketonemia or ketonuria or blood glucose levels >13.8 mmol/L HHS: indicated to bicarbonate levels of >18 mmol/L and plasma glucose levels of >33.3 mmol/L with or without the presence of ketonuria |
| [71] Bassili, Amal.et al., | 2001 | Egypt | Cross-sectional | 134 | T1 and T2 | | 46.3 | 16.7 | Incidence | ICD code |
| [70] Ndizihiwe, Eulade | 2021 | Tertiary hospitals Rwanda | prospective study | 143 | T1 and T2 | 59 | 35.6 | | Prevalence | ICD code |

(*Continued*)

**Table 1.** (Continued)

| Authors | year | Name of Developing countries | Study type | DMEs Sample size (n) | Population (DM type) T1 (T1DM) T2 (T2DM) | Magnitude of diabetic emergencies (%)[a] | | | Outcomes Data type (Prevalence/ Incidence) | Definitions of Diabetic emergencies |
|---|---|---|---|---|---|---|---|---|---|---|
| | | | | | | DKA | HHS | SH | | |
| [72] Ponesai, Net al., | 2015 | Chirumanzu, Zimbabwe | Case control | 68 | T1 and T2 | 44 | | 10.3 | Prevalence | DKA and HHS: A case of severe hyperglycemia was any patient with a fasting blood sugar >7.0mmol/L or RBG>11.1mmol/L, requiring admission for blood sugar control as determined by the clinician in charge SH: A case of hypoglycemia was any patient previously diagnosed with diabetes mellitus and put on treatment who had a blood sugar reading less than 3.5mmol/L |
| [76] Wu, X-y., et al., | 2020 | Shanghai, China | 5-year retrospective | 158 | T2 | 41.1 | 46.8 | | Prevalence | DKA: random blood glucose, (RBG) usually > 16.7 mmol/L), positive or strongly positive urine or blood ketone, and arterial PH < 7.30 or blood bicarbonate < 18 mmol/L HHS: (RBG > 33.3 mmol/L), venous PH > 7.25 or arterial PH > 7.30, blood bicarbonate > 18 mmol/L, AG < 12, strongly positive urine glucose, negative or weakly positive urine ketone, effective plasma osmolality > 320 mmol/L |

NB: [a]*the diabetic emergencies are presented as percentages; However, the sum of the percentages of each type of complication does not add up to 100%, as patients can have more than one acute complication, since these complications are not mutually exclusive*

lowest prevalence of HHS at 3.3%, while Nigeria reported a higher prevalence of 58% for HHS (59, 69). On the other hand, the prevalence of severe hypoglycemia was reported with in a range of (3.3%- 64.7%) per year in Ethiopia and Zimbabwe respectively [61, 72]. The incidence of diabetic emergencies varied across studies included in this review. As summarized in Table 1, the incidence of DKA ranged from 3.8% to 12.5% per 100 patients per year in studies conducted in China and Ethiopia, respectively. Hyperosmolar hyperglycemic state (HHS) showed a broader range, with incidences reported between 2.1% and 46.3% per 100 patients per year in Ethiopia and Egypt, respectively. Furthermore, the incidence of severe hypoglycemia ranged from 8.3% to 16.7% per 100 patients per year in studies from Mexico and Egypt.

## Risk factors of diabetic emergencies in developing countries

As shown below in (Fig 3) out of the twenty studies fifteen studies reported, infections were the most risk factors associated with the diabetic emergencies followed by noncompliance towards the medications they had taken and new onset of diabetes. Other precipitant factors

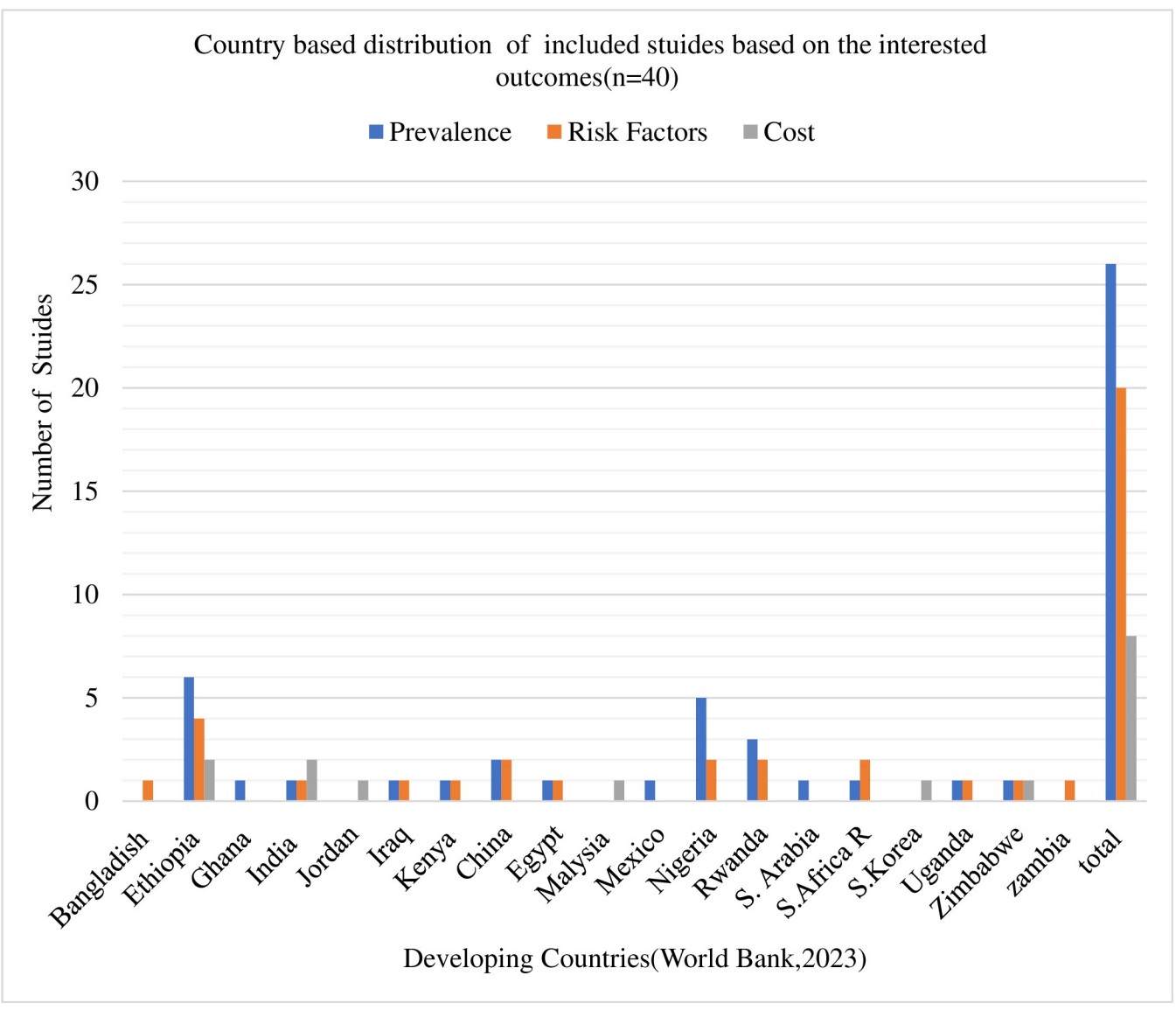

**Fig 2. Countries-based distributions of included studies.** This chart is showing the country-based distribution of included studies which describe the geographical representation of the data.

were also reported like missed meal, smoking, younger age, high distance form healthcare setting, uninsured patients and low educational status (S1 Table).

Although limited reports(n = 8), studies from Africa and Asian developing countries were included in this systematic review which reported the direct (hospitalization, medication and diagnosis) cost of diabetic emergencies Table 2. Reports for the costs of hospitalization that were associated with the DMEs events were taken from the patients' perspective, and the average direct economic cost of each DMEs event per patient was estimated. A report from Zimbabwe and Jordan has presented the lowest (105 USD/Patient/event) and highest (230 USD/Patient/event) costs respectively [79, 80]. Indirect costs such as productivity losses resulting from lost workdays and direct non-medical costs were not available in patients' records at the hospitals and thus, not considered. All costs were inflated to costs in 2023 and no studies extrapolated the cost estimates to the national population.

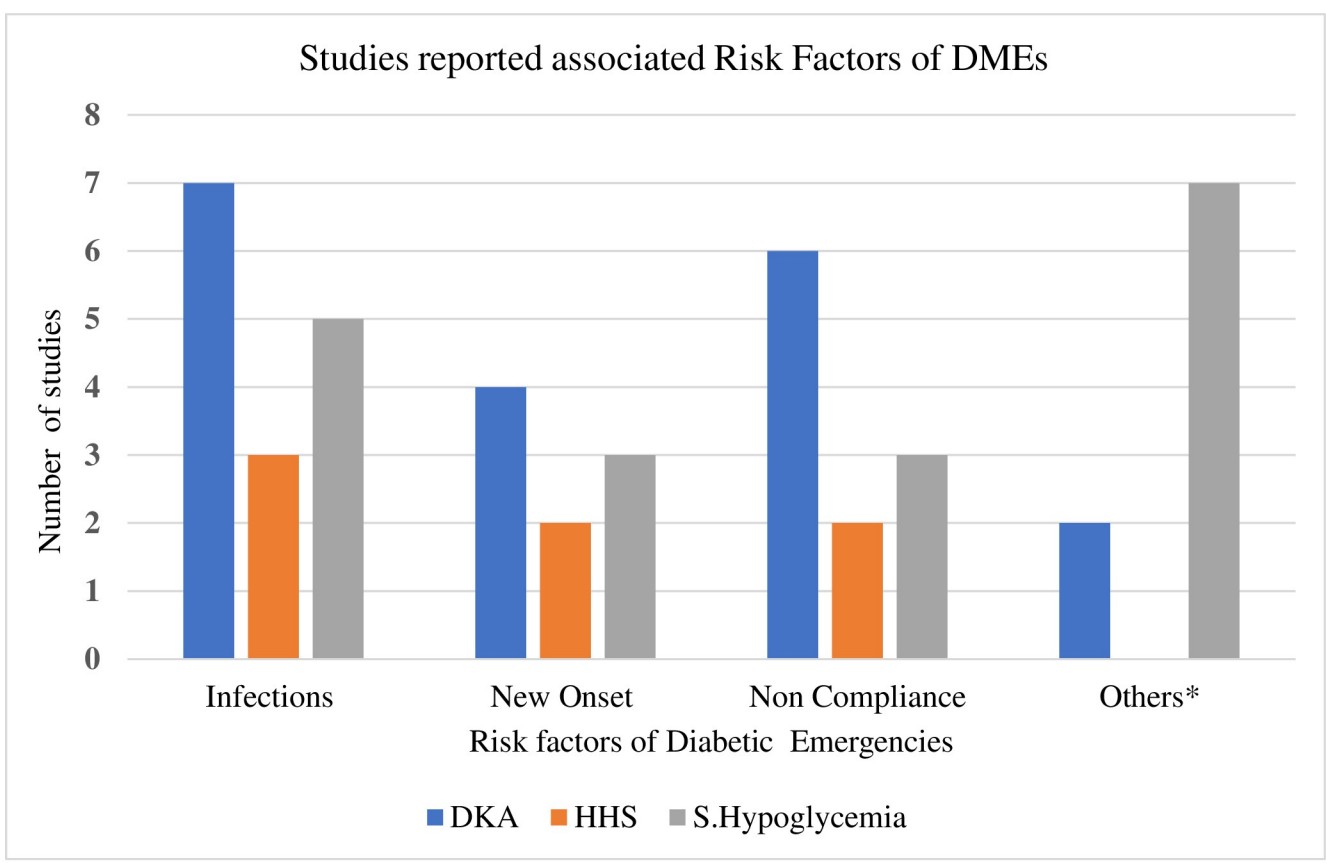

*Others risk factors: Socio demography characteristics of DM patients (Age difference, Duration of diabetic history, Exercise, Alcohol use etc.)*

**Fig 3. Number of studies associated risk factors of diabetic emergencies.**

**Table 2. Reported summary of the cost expenditure on diabetic emergencies (DMEs) in developing countries among diabetic patients.**

| Author | year | country | Study type | Cases | Type of DM (T1:T1DM, (T2:T2DM) | D. Cost of diabetic emergencies mean USD/Patient/event (2023USD) |
|---|---|---|---|---|---|---|
| [43] Assefa, B., et al., | 2014 | Ethiopia | Cros-sectional | 130 | T1and T2 | 125 |
| [44] Feleke, Y. and Enquselassie | 2007 | Ethiopia | Case control | 146 | T1and T2 | 154 |
| [79] Naser, A. Y. et al., | 2020 | Jordan | Cross- sectional | 126 | T1and T2 | 230 |
| [80] Mutsa P. Mutowo | 2016 | Zimbabwe | Cohort (Retrospective) | 64 | T2 | 105 |
| [81] Aljunid SM., et al., | 2019 | Malysia | Cross sectional | 244 | T2 | 183 |
| [82] Acharya, L. D. et al., | 2016 | India | Case control | 116 | T2 | 113 |
| [83] Kansra, P., and Oberoi, S., | 2023 | India | Cross-sectional | 89 | T1and T2 | 176 |
| [84] Ha WC., et al., | 2012 | South Korea | Cohort (Retrospective) | 320 | T1 and T2 | 133 |

Direct health care Cost: considers only for Expenditures of diagnosis, medicines and Hospitalization fee of the diabetic emergencies

**Quality of included studies.** Most studies presented and explained their results in a clear way, consistent with the methodology of the study. Presentation of results was generally in agreement with the study aim and the conclusions were made in line with the results presented. More than 80% of the included studies carefully described the epidemiological sources, activity data, and unit costs regarding the diabetic emergencies. A few studies (11.5%) included in this review reported on prevalence studied on T1DM patients and only one study included was studied on T2DM patients. In Table 3, the assessment of the Quality Index scores for the Cross-sectional studies included in the review revealed that nearly seven articles achieved a score exceeding half, meeting the criteria for inclusion in the analysis. Additionally, for the included cohort studies(n = 14), the Newcastle-Ottawa Quality Assessment Scale was used to assess the risk of bias (S2A Table). Besides, few studies did not discuss any limitations regarding their collective definition of DKA and HHS as hyperglycemic crisis which might need a separated definition to report the specific prevalence and cost estimations.

Common weakness discussed in most of the studies was that collected data from patient interviews were subject to recall and social desirability bias. Also, the use of one study site and small sample sizes meant that results were not applicable to other sites or national estimates. the proportion of studies that met the criteria for the reporting of the direct cost expenditure to diabetic emergencies used in this review. According to the key methodological questions asked for the studies(n = 8), the answer for Yes with a total score was 53(66%) in 10 questions asked indicating an adequate quality assessment for most of the studies (S2B Table). It should

**Table 3. Quality index score for cross-sectional studies included in the systematic review.**

| Quality *assessment* Questions | List of included studies | | | | | | | | | |
|---|---|---|---|---|---|---|---|---|---|---|
| | [73] Kasinathan (2013) | [62] Rudasingwa (2012) | [58] Tilaye (2021) | [74] Li J. 2014 | [67] Gebre (2019) | [78] Doubova (2018) | [71] Bassili (2001) | [61] Negera, (2020) | [75] Al-Obaidi (2019) | [64] Bateganya, (2003) |
| Were the criteria for inclusion in the sample clearly defined? | Y | Y | Y | Y | Y | Y | Y | Y | Y | Y |
| Were the study subjects and the setting described in detail? | U/C | N | Y | U/C | Y | U/C | U/C | Y | Y | N |
| Was the exposure measured in a valid and reliable way? | Y | U/C | U/C | Y | Y | N | Y | Y | Y | N/A |
| Were objective, standard criteria used for measurement of the condition? | Y | Y | Y | Y | U/C | Y | Y | Y | N | Y |
| Were confounding factors identified? | Y | U/C | N | Y | N/A | Y | N | N | Y | N |
| Were strategies to deal with confounding factors stated? | U/C | Y | Y | N/A | Y | Y | Y | N | Y | Y |
| Were the outcomes measured in a valid and reliable way? | Y | Y | Y | Y | N | Y | Y | Y | N | Y |
| Was appropriate statistical analysis used? | Y | Y | N | Y | Y | Y | N | Y | N | U/C |
| *TOTAL score* | | | | | | | | | | |
| *Yes(Y)* | 6 | 5 | 5 | 6 | 5 | 6 | 5 | 6 | 5 | 4 |
| *No(N)* | 0 | 1 | 1 | 0 | 1 | 1 | 2 | 2 | 3 | 2 |
| *Un Clear (U/C)* | 2 | 2 | 2 | 1 | 1 | 1 | 1 | 0 | 0 | 1 |
| *Not Applicable(N/A)* | 0 | 0 | 0 | 1 | 1 | 0 | 0 | 0 | 0 | 1 |

be noted that the checklist has been adapted to the needs of this review and questions were benchmarked against the objectives of the study under review. A few studies did not clearly articulate or provide explicit information regarding the methodologies followed [43, 79].

## Discussion

Developing countries are faced with challenges in addressing the rise in NCDs whilst still grappling with infectious diseases. This threatens to overwhelm an already overstretched healthcare sector and pose a challenge to economic development these resource limed nations. As the countries move towards health coverage of diabetes mullitus, robust estimates of the magnitude, economic expenditure and the factors associated with diabetes related complications can be used to forecast the financial needs and the health care service utilization at global and national level. There is thus growing need to assess the burden associated with these DMEs in developing countries and prioritize interventions that prevent or delay the occurrence of these events. Although, estimates of costs of DMEs, their respective prevalence as well as risk factors have been published sufficiently in the developed world [85, 86] obtaining more valid and reliable data of DMEs in resource limited setting is still very difficult. Lack of testing facilities and equipment, inadequate number of trained health personnel, poor access to health facilities and lack of awareness are some of the barriers [87, 88].

### Magnitude of diabetic emergencies

This is the first comprehensive review focused on the magnitude, risk factors and economic impact of DMEs in the developing countries which may contribute to understanding of the range of DMEs, prevention of precipitant factors and estimation of the direct costs' expenditure for the events of the complications in the resource limited health care setting. The first objective of this review was to measure the magnitude and capture the evidence presented in the literatures regarding the DMEs that have been published since 2000. The literature search identified diabatic emergencies prevalence related 26 studies from Africa(n = 20), Asia(n = 5) and Mexico (n = 1) that met the eligibility criteria of this review. Our findings suggested that, despites their magnitude were huge, DMEs prevalence were varied inter and intra of the developing countries. The variation in prevalence rates among the studies may reflect differences in the study populations, such as patients with diabetes mellitus versus those with DKA, HHS, or hypoglycemia. As such, these findings highlight the need to consider the specific population and disease context when interpreting prevalence data.

The compiled finding of this of systematic review showed that DKA is the predominate diabetic emergencies among younger age and T1DM diabetic patients ranging 6.7% in Uganda to 83% in Saudi Arabia [64, 77]. Conforming with this, Studies from the developed countries across Europe, Australia, New Zealand, and the United States were reported a mean prevalence of DKA 29.9%, ranging from 19.5% in Sweden to 43.8% in Luxembourg [89]. Reports from both sides suggested that younger children are at the greatest risk for presentation of DKA at T1DM diagnosis because the symptoms of T1DM may go unrecognized and only present for medical attention after development of complications [59, 90]. On the other hand., in a systematic literature review conducted in 2017, encompassing 19 studies across Europe (n = 5), North America (n = 11), Israel (n = 2), and China (n = 1), the incidence of diabetic ketoacidosis (DKA) varied, with Sweden recording the lowest prevalence at 0.0%, while Canada reported the highest prevalence at 12.8% [91].

Many studies included in our review note that the reason for such prevalence difference is accessibility of diabetes technologies like continuous glucose monitoring (CGM) protocols and better awareness towards the diseases complications in the developed countries could play

a positive role on reducing the prevalence of DKA in relative to the developing countries. Many of the developing countries represented in this review, access to glucose monitoring devices and medications is limited. For example, in Ethiopia, less than 30% of patients with diabetes have access to glucometers, and even fewer have access to ketone testing tools [92]. Similarly, in Zimbabwe, financial constraints significantly hinder access to self-monitoring devices, with a substantial number of patients unable to perform routine glucose checks [93]. Access to diabetes medications is another critical barrier. Studies indicate that only 50–60% of individuals in developing countries can afford essential treatments such as insulin due to high costs and inconsistent supply chains [94, 95]. This lack of resources significantly affects the ability to prevent or manage emergencies like diabetic ketoacidosis (DKA) and hyperosmolar hyperglycemic state (HHS), particularly in resource-limited settings.

In line with our findings, Evidence from single-center studies suggest that DKA in new-onset T1DM is more common in developing countries compared to developed countries, with rates ranging from 62.2 to 77.1% in Nigeria, 69.8% in South Africa, and 92.1% in Sudan [96–98]. In this review, although some studies discussed the coexistence of HHS and DKA as a hyperglycemic crisis, the majority highlighted a higher prevalence in T2DM patients in developing countries. These diabetic emergencies, continue to pose a significant threat to the mortality and morbidity of diabetic patients. Efforts are needed to address and prevent the escalating burden on patients.

Additionally, the review examined the prevalence of another diabetic emergency, severe hypoglycemia. This condition poses a substantial burden, contrasting with DKA and HHS, primarily linked to insulin-dependent diabetes mellitus. Its occurrence tends to increase as metabolic control approaches below normal levels. In line with our finding a study from Atlanta reported about 0.5% prevalence of severe hypoglycemia. While severe hypoglycemia is prevalent in children and younger adults, the rising incidence of HHS among this demographic is remarkable like a report in USA with a prevalence range of 4.4%- 42.9%. This increase is attributed to the exponential growth of obesity and the escalating cases of T2DM in this age group [11]. Generally, Acute complications exhibit a similar trend, with an overall prevalence of 30.5%. Among these, DKA is the most prominent at 71%, followed by hypoglycemia at 19.4%. In contrast, the occurrence of HHS is relatively insignificant [59, 98].

## Factors associated with diabetic emergencies

Our second objective was to review the factors associated with diabetic emergencies in the developing countries. like other studies, most (n = 15) of the eligible studies which include in this review reported that infection is the most common precipitating factor in the development of DKA and HHS [99–103]. In consistent with this review a large survey in UK, find that infection was identified as the most common precipitating factor for diabetic ketoacidosis (45%), followed by insulin omission (20%); other causes included newly diagnosed diabetes and alcohol or drug related problems [104]. The IDF 2015 report verified that about 66.7% of people with diabetes in Africa are assumed to be undiagnosed. This unmet need for diabetes diagnosis is a result of weak health systems in many developing countries that fail to screen patients for diabetes and make them hospitalized and newly diagenesis because of the diabetic emergencies. Specifically, precipitating factors associated with the development of DKA were identified in many findings and the reports we include, with the commonest being infection and treatment non-compliance [104–106]. As controlling infections is a challenge in resource limited health care settings of developing countries, its precipitating effect on DMEs becomes worst. Besides, a worldwide wide report by Wabe et al., stated that adherence rate on medications for diabetes control vary between 36% and 93%. Furthermore, finding from Nigeria and Ethiopia

with reported that more than 50% of T2DM were non-adherent to the prescribed anti-diabetic drugs due to lack of finance to purchase which made theme unable to procure insulin and other medications required for their treatments [42, 59, 107].

Additionally, a retrospective case-control study in US reported a clear association between readmission for DKA or HHS and young age (< 35 years), behavioral health comorbidity like depression), self-pay or no insurance status, and drug and alcohol abuse which support the findings in our review [16]. Decreased patient experience with diabetes management, developmental issues, an end of parental oversight and more irregular living arrangements may be associated with an increased risk for the diabetic emergences of younger age. Self-pay due to limiting access to healthcare as well as drug and alcohol abuse and behavioral health problems like depression, in turn, may exert their effects by interfering with treatment engagement, self-efficacy, and self-care. Thus, verification of the precipitant factors of the diabetic emergencies might suggests potential objective predictors of the events recurrence that could assist health care professionals to prevent readmission and hospital management. On top of that, consideration of these recognizable risk factors has very relevant input to improve patient care and to reduce healthcare costs. Reports from Iran showed that Patient education and regular supply of insulin from a tertiary center appeared to have a protective effect against DKA. A part of patient education is the correct insulin administration which has been found also in another study to reduce the DKA risk [72, 75].

## Economic impact of diabetic emergencies

A significant body of evidence shows that the growing epidemics of diabetes and associated diabetic complications worldwide poses catastrophic financial costs, especially in developing countries where most of the expenses are paid by patients and families. The estimated costs for diabetic emergencies per individual vary significantly across the reviewed studies, reflecting disparities in economic contexts and healthcare systems. According to the world data, one of the reviewed studies from Ethiopia, the cost is estimated at $154, which represents nearly 14% of the average annual income ($1,140), placing a significant financial strain on individuals and households. Similarly, In Zimbabwe, the cost expenditure for diabetic emergencies is almost $105, constituting approximately 8% of the average annual income ($1,300), which still a substantial burden in a resource-limited setting. Conversely, in the study from South Korea showed that the cost expenditure for diabetic emergencies has reported about $133, which is only 0.27% of the average annual income ($48,922), highlighting the reduced economic burden in high-income nations.Few identified studies examined the economic implications of diabetic emergencies, and most were limited to examination of medical resources, but none provided episode-related costs that may be considered in health economic evaluations to compare alternative treatment strategies that may reduce or prevent the incidence of the complications [108]. These findings emphasize the disproportionate economic impact of diabetic emergencies in low- and middle-income countries, where healthcare costs constitute a higher percentage of household income, further straining already limited resources. Addressing these disparities requires strategic interventions to improve diabetes management and reduce emergency events.

As third objective, this study aids to fill the gap of economic data estimations informing the consequences of these emergency episodes in resource limited health care settings. In a recent comprehensive literature review examining the economic impact of diabetic ketoacidosis (DKA) in pediatric and adolescent populations, particularly among users of insulin pumps, Varughese highlighted that the burden is significant for both patients and healthcare systems, necessitating considerable global resource allocation [109]. Similarly, one of the studies we

included in this review reported that patient with severe hypoglycemia incur about $712 (52%) more in costs per month after their hospitalization than on comparison with other patients without the complication [35].

Although only eight studies are presented regarding the economic expenditure of diabetic emergencies in this review, all studies provided a direct cost (hospitalization, diagnosis and treatment) expenditure for the diabetic emergencies like most of the existing studies, in resource limited nations. On the other hand, reports from the developed world like a study in US reported about $972-$1,499 direct cost expenditure for diabetes emergencies related cases [110]. This difference could suggest us the health care service utilization (accessibility of medications and diagnosis instruments) standard difference between the developed and resource limited countries quite magnificent. The estimations of costs of diabetic emergencies in many developing countries may be underestimated until now due to absence of data on the relative contribution of cost of diabetes complications which needs more investigation on the constraint issues.

We reviewed the available evidences in the resource limited countries regarding the economic impact of diabetic emergencies among known diabetic patients. Despite heterogeneity across studies in terms of antidiabetic medications used, length of hospital stays, geography, patients' clinical and demographic characteristics, most findings were consistent [43, 44, 81, 82, 84].

Similar to the studies we incorporated, the majority of existing research in Africa focused solely on estimating direct costs. However, majority of the studies we included were hospital-based studies involving small sample sizes; as such results might not be applicable to national estimates. Likewise, a major limitation of most studies is that sensitivity analysis was not performed. The analyzed studies employed diverse methodologies to estimate costs, making it challenging to simplify or compare results across different studies. Generally, from this review it is clear that national estimates on the economic impact of DMEs in developing countries are lacking. Therefore, this review provides a fragmented picture of the economic impact of these complications in developing countries. As a result, it is highly likely that the aggregate costs associated with DMEs have been severely under estimated. Particular studies that provide national estimates of unmet diabetes acute complications are important for carefully estimating the national impact of the disease in general.

## Limitations

The current review has some limitations. The exclusion of articles not written in English could have led to the omission of relevant articles in the area under study. The review specifically concentrated on peer-reviewed articles and omitted grey literature, such as academic theses. This exclusion could introduce bias into the review, and conducting a broader search without this restriction might yield different outcomes. Similar to many other checklists, the results presented in the checklist may not be readily replicable as they represent a subjective assessment by the reviewer. Furthermore, the heterogeneity in study designs and cost estimation methods used makes the costs presented in the study non-comparable. As a result, it was not possible to conduct a meta-analysis. Another common weakness discussed was that studies collected data from single setting or insufficient sample size, thus disallowing the findings to be extrapolated for national estimates.

Studies that included in this review are identifying predictors of diabetic emergencies; however, they have been in mixed populations of patients with T1DM and T2DM. Thus, reporting separately in children and adolescents, or in other highly selected populations may limit their generalizability.

## Conclusion

In conclusion, the systematic review's findings underscore the pressing need for proactive measures to address the high prevalence of diabetic emergencies in developing countries. Moreover, infections, non-compliance, and recent onset of diabetes stand out as the leading factors triggering diabetic emergencies. Recognizing individuals at greater risk of diabetic emergencies is crucial for healthcare providers to implement tailored strategies, ultimately reducing emergency visits and hospital admissions. Moreover, the economic implications revealed in the review emphasize the substantial burden diabetic emergencies impose on national healthcare systems in developing countries. Consequently, integrating preventive measures into diabetes management programs is imperative, not only for the well-being of affected individuals but also to alleviate the significant strain on limited health resources and expenditures in these nations.

## Supporting information

**S1 Table. Description of qualitative analysis of included studies.**
(PDF)

**S2 Table.** A: Risk of bias assessment of the studies included in this systematic review, B: Methodological Quality of the economic evaluation studies.
(ZIP)

**S1 File. Descriptions of 548 excluded records and reasons.**
(XLS)

**S2 File. PRISMA checklist report.**
(PDF)

## Acknowledgments

We extend our sincere appreciation for the support and mentorship provided by the Addis Ababa University School of Pharmacy throughout the research process as well as individuals and organizations who contributed to the realization of this systematic review.

## Author Contributions

**Conceptualization:** Halefom Kahsay Haile, Teferi Gedif Fenta.

**Methodology:** Halefom Kahsay Haile, Teferi Gedif Fenta.

**Supervision:** Teferi Gedif Fenta.

**Writing – original draft:** Halefom Kahsay Haile.

**Writing – review & editing:** Halefom Kahsay Haile.

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
