## [Decision Letter · Decision Letter 0]

14 Oct 2024

PONE-D-24-05539Magnitude, risk factors and economic impacts of diabetic emergencies in developing countries: A systematic reviewPLOS ONE

Dear Dr. Haile,

Thank you for submitting your manuscript to PLOS ONE. After careful consideration, we feel that it has merit but does not fully meet PLOS ONE’s publication criteria as it currently stands. Two external referees identify merit and potential within the manuscript, but raised several concerns within the manuscript. Therefore, we invite you to submit a revised version of the manuscript that addresses the points raised during the review process.

We look forward to receiving your revised manuscript.

Kind regards,

Neftali Eduardo Antonio-Villa, MD PhD

Academic Editor

PLOS ONE

Journal Requirements:

2. As required by our policy on Data Availability, please ensure your manuscript or supplementary information includes the following: 

5. Please ensure that you refer to Figure 1 and 2 in your text as, if accepted, production will need this reference to link the reader to the figure.

Reviewers' comments:

Reviewer's Responses to Questions

**Comments to the Author**

1. Is the manuscript technically sound, and do the data support the conclusions?

Reviewer #1: No

Reviewer #2: No

2. Has the statistical analysis been performed appropriately and rigorously? 

Reviewer #1: No

Reviewer #2: Yes

3. Have the authors made all data underlying the findings in their manuscript fully available?

Reviewer #1: No

Reviewer #2: Yes

4. Is the manuscript presented in an intelligible fashion and written in standard English?

Reviewer #1: No

Reviewer #2: No

5. Review Comments to the Author

Reviewer #1: Thank you for submitting interesting paper.

I have the folliwing concerns.

1. Abstract

・Some words are misspelled. For example, sever is severe, and Diabetes Mullites is Diabetes Mellitus.

2. Introduction, Data Sources and search strategies

「We searched PubMed, COCRANE, MEDLINE, EMBASE…」

・COCRANE should be Cochrane. Make sure it is not a typo.

3. Materials and methods, Descriptions of the interested outcomes.

・What does the word “RBGL” mean? Please spell it in full when it first appears.

4. Results Prevalence of diabetic emergencies]

「Studies that carried out in China and Ethiopia has documented a range of prevalence rates or DKA episodes, from 3.8% to 73.4%, respectively (61, 74).」

4-1

・has→have

Studies that carried out in China and Ethiopia have documented a range of prevalence rates or DKA episodes, from 3.8% to 73.4%, respectively (61, 74).

4-2

・“The prevalence of diabetic ketoacidosis (DKA) in the study by Ahmed MM et al., referenced as ref 77, was 83%.I think the ratio will be 3.8% to 83%.

5. Table 1

5-1

・”Sever Hypoglycemia” should be “Severe Hypoglycemia”.

5-2

・Cross sectional is described in multiple ways, so please unify.

5-3

The items I noticed are only a small part of the critical errors. The author should thoroughly review and correct the entire content.

・The studies included in this table have such varying definitions that they cannot be compared. Particularly, prevalence should be reconsidered.

〇Prevalence and Incidence

For example,

・The ref 59 Edo study, focuses on prevalence(58%).

・The ref.69 Abate study, focuses on incidence(0.9%).

・The ref.61 Negera study likely focuses on prevalence（73.4%).

・The ref.74 Li J study, on the other hand, deals with incidence(3.8%).

〇Different definition of Prevalence

・The ref.67 Gebre B study explores the prevalence of diabetes complications among patients with diabetes mellitus.The 61 Negera study specifically looks at the prevalence of diabetes complications among patients with DKA, HHS, and hypoglycemia.

・The ref.20 SarfoKantanka study focuses on the prevalence of diabetes complications among patients with DKA, HHS, hyperglycemia and hypoglycemia.

・However, the ref.77 Ahmed study focuses on the prevalence of diabetes complications among patients with DKA and HHS.

Therefore, these studies are not directly comparable due to their different approaches and definition. If the overall number is defined as different, it is natural that it is not possible to compare the diseases contained in it by proportion.

6. Discussion, Prevalence of diabetic emergencies

「The review encompassed 26 studies from Africa (n=14), Asia (n=5),and Mexico (n=1), revealing variations in DME prevalence both within and between countries in　the developing world, underscoring the considerable scope of the issue.」

・Wasn’t the number of the studies from Africa 20?

That's all for my comments. Thank you.

Reviewer #2: The study is a systematic review of diabetic emergencies in developing countries. In total around 40 studies were included, the majority from Africa. Assessments of prevalence of emergencies and costs are presented, and a large variability is noted on study outcomes between the studies. The study adresses an important topic, but should be revised regarding language, structure and interpretation.

Major:

1. The study uses an unspecific language with frequent bold statements without support from real numbers eg. ”significant”, ”high incidence”, ”increased”, ”hazardous implications”, ”inadequate”. All such statements should be revised with quantified evidence from the references used in these sentences. A good example where this is already done is in the first paragraph of the introduction ”lower overall disease burden by 5-10%”.

2. With respect to the research question of the study, it may not be fair to include studies from developing countries across the globe, as there may be substantial differences between different regions. A subgroupanalysis only including countries in a specific region could resolve this problem. This should at least be done examining outcomes only examining studies from Africa, where most studies were performed. Potential regional differences should also be further discussed in the study.

3. The current conclusion in the abstract is not drawn from the results in the study and must be revised (the conclusion of the manuscript is fair). Please consider who is the intended reader of this manuscript, descision makers? The large variability between studies prevents any strategic conclusion on policymaking. Which new studies should be conducted, and where? What interventions could reduce emergencies, and how should these interventions be tested? What does the estimated costs mean for the current treatment strategies, would it be cheaper to treat diabetes more effectively? These aspecst are currently not discussed sufficiently.

Minor:

1. There are some places where data is unclear. Eg the summary of the 26 studies is detailed but only adds up to 20 studies. All numbers must be doubble checked.

2. The cost of emergencies is estimated around 200 dollars. This is a quite normal anual medication cost for any patient in a western country, and is from this perspective not considered high. Therefore, some context must be provided for this number to make any sense. What is the annual income in these countries, what proportion of healthcare costs does this constitute?

3. Hard numbers on the diabetesrelated mortalities in developing countries could further support the importance of this study.

4. Information regarding the current state of diabetes treatment in these regions is not discussed. CGM is meantioned, but is the mesuring of glucose in patients available at all for these individuals? What proportion have access to diabetes medications, and what proportion can measure glucose or ketones them selves in any way?

6. PLOS authors have the option to publish the peer review history of their article (what does this mean?). If published, this will include your full peer review and any attached files.

Reviewer #1: No

Reviewer #2: No

---

## [Author Response · Author response to Decision Letter 0]

11 Dec 2024

Dear Editor!

 for your concern, comments and suggestions in raised in 1-3, We addressed and prepared our manuscript based on the journal guidelines and format and specifically 

 Response: Revised and amended

5. Please ensure that you refer to Figures 1 and 2 in your text, as, if accepted, production will need this reference to link the reader to the figure.

Thank you for your guidance. I have carefully reviewed the manuscript and ensured that Figures 1 and 2 are explicitly referenced in the text. Specifically:

• Figure 1, which illustrates the Preferred Reporting Items for Systematic Reviews and Meta-Analyses (PRISMA) flow chart for study identification, screening, and inclusion, has been referenced where I discuss the study selection process.

• Figure 2, showing the country-based distribution of included studies, is cited when describing the geographical representation of the data.

Dear Reviewers!

Thank you for taking the time to review our manuscript titled "Magnitude, risk factors, and economic impacts of diabetic emergencies in developing countries: A systematic review." We sincerely appreciate your thoughtful feedback and constructive suggestions. We have carefully reviewed your comments and made the necessary revisions to strengthen our manuscript. and we attached a word file titled Response to Reviewers, which responded to both reviewers that we provide detailed responses to each point raised. 

Regards, 

Halefom Kahsay

---

## [Decision Letter · Decision Letter 1]

22 Dec 2024

PONE-D-24-05539R1Magnitude, risk factors and economic impacts of diabetic emergencies in developing countries: A systematic reviewPLOS ONE

Dear Dr. Haile,

Thank you for submitting your manuscript to PLOS ONE. After careful consideration, we feel it has merit and may be suitable for publication after considering minor revisions from its current form. Therefore, we invite you to submit a revised version of the manuscript that addresses the points raised during the review process.

We look forward to receiving your revised manuscript.

Kind regards,

Neftali Eduardo Antonio-Villa, MD PhD

Academic Editor

PLOS ONE

Journal Requirements:

Additional Editor Comments:

Dear authors, please make the following changes to the manuscript, particularly those enlisted by referee 2.

Here are some of my specific considerations: 

1. Across all text, consider using the term “adults living with diabetes” instead of "adults with DM" for clarity and consistency.

2. In the abstract, correct the typo in the phrase "Sever hypoglycemia" to "Severe hypoglycemia."

3. In the introduction section, in the line “Non-communicable diseases (NCDs) are expected to increase and surpass infectious diseases by 2030, according to predictions," specify the source of predictions—are they from the NCD risk collaboration or the GBD studies?

4. Conduct a thorough check for typos and extra spaces throughout the manuscript. Examples include “Diabetes Mullites” and “irrelevant conte," as well as "sever hypoglycemia."

5. Please revise and ensure that the citations adhere to PLOS ONE guidelines, as some citations do not include the journal name or the year of publication.

Reviewers' comments:

Reviewer's Responses to Questions

**Comments to the Author**

1. If the authors have adequately addressed your comments raised in a previous round of review and you feel that this manuscript is now acceptable for publication, you may indicate that here to bypass the “Comments to the Author” section, enter your conflict of interest statement in the “Confidential to Editor” section, and submit your "Accept" recommendation.

Reviewer #1: All comments have been addressed

Reviewer #2: All comments have been addressed

2. Is the manuscript technically sound, and do the data support the conclusions?

Reviewer #1: Yes

Reviewer #2: Yes

3. Has the statistical analysis been performed appropriately and rigorously? 

Reviewer #1: Yes

Reviewer #2: Yes

4. Have the authors made all data underlying the findings in their manuscript fully available?

Reviewer #1: Yes

Reviewer #2: No

5. Is the manuscript presented in an intelligible fashion and written in standard English?

Reviewer #1: Yes

Reviewer #2: Yes

6. Review Comments to the Author

Reviewer #1: Thank you for responding to our comments.

The authors performed a systematic review to assess the magnitude, risk factors and economic impacts of diabetic emergencies in developing countries.

I think it has improved significantly after the initial revisions

I have several concerns.

【Minor comments】

1)(Introduction)

・The global incidence of diabetes is rising rapidly, posing a significant public health challenge, particularly in developing countries, which account for approximately 80% of the 537 million people living with diabetes worldwide as of 2021.

〇Please add the reference of the facts.

2)(Introduction)

〇The sentence 「it characterized by abnormally high or low blood sugar levels, are significant contributors to morbidity, mortality, and increased healthcare costs among adults with diabetes. For instance, diabetic ketoacidosis (DKA) and hyperosmolar hyperglycemic state (HHS) collectively account for up to 30% of diabetes-related hospital admissions in developing countries, with mortality rates ranging from 5% to 20% depending on the region and healthcare access.」 may be 「it is characterized by abnormally high or low blood sugar levels, are significant contributors to morbidity, mortality, and increased healthcare costs among adults with diabetes. For instance, diabetic ketoacidosis (DKA) and hyperosmolar hyperglycemic state (HHS) collectively account for up to 30% of diabetes-related hospital admissions in developing countries, with mortality rates ranging from 5% to 20% depending on the region and healthcare access.」.

3)(Introduction)

〇The sentence「Even though diabetic emergencies are becoming more common in poor nations, there is a conspicuous Visible scarcity of paucity of thorough in-depth studies in this area, and the research that is currently available lacks statistical clarity and specificity.」may be [Even though diabetic emergencies are becoming more common in poor nations, there is a conspicuous visible scarcity of paucity of thorough in-depth studies in this area, and the research that is currently available lacks statistical clarity and specificity.].

4)(Results, Table 1)

The authors are calculating the prevalence, but the denominators differ.

・The ref.67 Gebre B study explores the prevalence of diabetes complications “among patients with diabetes mellitus.” The denominators are patients with diabetes mellitus.

・

However, the 61 Negera study specifically looks at the prevalence of diabetes complications “among patients with DKA, HHS, and hypoglycemia”. The denominators are not the patients with diabetes, but rather the patients with DKA, HHS, and hypoglycemia.

・The ref.20 SarfoKantanka study focuses on the prevalence of diabetes complications “among patients with DKA, HHS, hyperglycemia and hypoglycemia”. The denominators are not the patients with diabetes, but rather the patients with DKA, HHS, hyperglycemia, and hypoglycemia.

→〇If the groups of diseases included as subjects differ in each study, then each prevalence cannot be compared.

5)(Results, Table 4)

〇Please fill in the blanks in the Total score 'Yes (Y).'

6)Overall

〇Please review the document overall for minor English mistakes.

Example:

Descriptions of the Interested Outcomes

・Severe Hypoglycemia → Severe Hypoglycemia.

Reviewer #2: I think the manuscript is acceptable for publication. There are some minor spelling-errors/formatting errors which should be addressed before publication.

7. PLOS authors have the option to publish the peer review history of their article (what does this mean?). If published, this will include your full peer review and any attached files.

Reviewer #1: No

Reviewer #2: No

---

## [Author Response · Author response to Decision Letter 1]

27 Dec 2024

Response document attached separately for reviewers as part of ''Response to Reviewers''

---

## [Editor Report · Decision Letter 2]

3 Jan 2025

Magnitude, risk factors and economic impacts of diabetic emergencies in developing countries: A systematic review

PONE-D-24-05539R2

Dear Dr. Haile,

We’re pleased to inform you that your manuscript has been judged scientifically suitable for publication and will be formally accepted for publication once it meets all outstanding technical requirements.

Kind regards,

Neftali Eduardo Antonio-Villa, MD PhD

Academic Editor

PLOS ONE

Additional Editor Comments (optional):

I want to congratulate the authors for their effort in addressing the reviewers' comments. All the considerations have been appropriately addressed, and I can proceed to recommend this manuscript for acceptance. 
---

## [Editor Report · Acceptance letter]

23 Jan 2025

PONE-D-24-05539R2 

PLOS ONE

Dear Dr. Haile, 

I'm pleased to inform you that your manuscript has been deemed suitable for publication in PLOS ONE. Congratulations! Your manuscript is now being handed over to our production team.

Kind regards, 

on behalf of

Dr. Neftali Eduardo Antonio-Villa 

Academic Editor

PLOS ONE